# Antimicrobial Treatment Options for Multidrug Resistant Gram-Negative Pathogens in Bone and Joint Infections

**DOI:** 10.3390/pathogens14020130

**Published:** 2025-02-01

**Authors:** Maria Tsilika, Fotinie Ntziora, Efthymia Giannitsioti

**Affiliations:** 11st Department of Internal Medicine, Medical School, National and Kapodistrian University of Athens, Laiko General Hospital, 11527 Athens, Greece; martsili@yahoo.gr; 21st Department of Propaedeutic and Internal Medicine, Medical School, National and Kapodistrian University of Athens, Laiko General Hospital, 11527 Athens, Greece; fotinientziora@gmail.com

**Keywords:** multidrug resistant, Gram-negative bacteria, bone and joint infections, antibiotics

## Abstract

Multidrug (MDR) and extensive drug (XDR) resistance in Gram-negative bacteria (GNB) emerges worldwide. Although bone and joint infections are mostly caused by Gram-positive bacteria, mainly Staphylococci, MDR GNB substantially increase also as a complication of hospitalization and previous antibiotic administration. This narrative review analyzes the epidemiological trend, current experimental data, and clinical experience with available therapeutic options for the difficult to treat (DTR) GNB implicated in bone and joint infections with or without orthopedic implants. The radical debridement and removal of the implant is adequate therapy for most cases, along with prompt and prolonged combined antimicrobial treatment by older and novel antibiotics. Current research and clinical data suggest that fluoroquinolones well penetrate bone tissue and are associated with improved outcomes in DTR GNB; if not available, carbapenems can be used in cases of MDR GNB. For XDR GNB, colistin, fosfomycin, tigecycline, and novel β-lactam/β-lactamase inhibitors can be initiated as combination schemas in intravenous administration, along with local elution from impregnated spacers. However, current data are scarce and large multicenter studies are mandatory in the field.

## 1. Introduction

Antimicrobial resistance is an emerging issue globally. It is estimated that 4.95 × 10^6^ deaths are attributed to resistant pathogens, of which more than 100,000 are due to methicillin-resistant *Staphylococcus aureus* (MRSA) and 50,000–100,000 are due to difficult to treat (DTR) pathogens able to “escape” the effect of antibacterial drugs, such as *Enterococcus faecium*, *Klebsiella* spp., *Acinetobacter baumannii*, *Pseudomonas aeruginosa*, and *Enterobacter* spp., which are also responsible for more than 2/3 of healthcare-associated infections [1]. Resistant pathogens are divided into three main groups: multi-drug resistant (MDR) if non-susceptible to at least one agent in ≥3 antimicrobial categories, extensively drug resistant (XDR) if non-susceptible to at least one agent in all but ≤2 antimicrobial categories, and pan drug resistant (PDR) if non-susceptible to all agents in all antimicrobial categories [2]. Prosthetic joint infections (PJIs) are an important complication in orthopedic surgery and are associated with increased length of hospitalization, the need for revision arthroplasties, higher healthcare cost, and mortality. Recent studies support that MDR pathogens play an important role in osteoarticular (OA) bone and joint infections (BJIs), the majority of which are attributed to Gram-positive bacteria (GPB), including resistant pathogens. As MDR GNB’s rate in PJIs is increasing, it is important to design successful treatments, making effective antibiotic stewardship at the same time [3]. MDR range from 5 to 30%, depending on the region and the site of infection [4,5,6,7]. *Klebsiella pneumonia* carbapenemases (KPCs) are most commonly seen in North America, Asia, and Southern Europe, New Delhi metallo-β-lactamase (NDM) predominates on the Indian subcontinent, the Middle East, and the Balkans, and Imipenemase (IMP) is seen in Asia and Australia [8].

We sought to review the available evidence on emerging MDR GNB and summarize laboratory and clinical data on antimicrobial options in BJIs. We performed a thorough search by relevant key words in PubMed: “bone and joint infections” AND “multidrug resistant” AND “Gram negative bacteria”. In addition, our search was specified by the type of BJI (“osteomyelitis”, “septic arthritis”, “prosthetic joint infection”, “ fracture related infection”, “ osteosynthesis”) by bacterial species and resistance mechanisms (e.g., ”*Pseudomonas aeruginosa*”, “Klebsiela KPC”, “carbapenem resistant Acinetobacter”, “*Esherichia coli*”, “ *Enterobacter* spp.” “ ESBL”, “fluoroquinolone resistant”) and by the type of resistance (“multidrug resistant”, “extensively drug resistant”, “difficult to treat”). Furthermore, some additional data were also retrieved by the references of papers relevant to experimental, epidemiological, and clinical data.

## 2. Epidemiology of Resistance of Gram-Negative Bacteria in Bone and Joint Infections

The epidemiology of GNB in PJIs varies depending on the time of the infection’s onset. In early PJI (within 3 months of prosthesis implantation), GNB, particularly Enterobacteriaceae, are more prevalent, representing 21.6% of cases. In late acute PJI (over a year following surgery with symptoms lasting less than 4 weeks and a seeding from an obvious source), the prevalence of Enterobacteriaceae is even higher at 28.8%. However, in delayed PJI (3–12 months after implantation) and late chronic PJI (over a year following surgery with symptoms lasting more than 4 weeks and no seeding from an obvious source), the prevalence of Enterobacteriaceae drops significantly to 5.4% and 3.8%, respectively. Non-fermenting GNB, such as *Pseudomonas aeruginosa*, have a low and consistent prevalence across all time periods, ranging from 0% to 4% [9].

Resistant GNB, such as those producing extended-spectrum beta-lactamases (ESBLs) or carbapenem-resistant Enterobacteriaceae (CRE), can complicate the treatment of PJIs. These resistant strains are often associated with healthcare settings and can be more challenging to treat due to limited antibiotic options. In a multicenter cohort study by Benito et al., it was shown that most infections are caused by Staphylococci, although the rate of infection by GNB and fungi increased from 2003 to 2012, as did the proportion of MDR infections, mainly due to the increase in resistant GNB. Multidrug-resistant GNB increased from 5.3% in 2003–2004 to 8.2% in 2011–2012 (*p =* 0.032); specifically, there was an increase over time in the proportion of MDR *Escherichia coli* (the proportion doubled from 2% in 2003–2004 to 4.3% in 2011–2012; *p =* 0.061), *Klebsiella pneumoniae* (0% in 2003–2004 to 1.1% in 2011–2012; *p =* 0.051), *Pseudomonas aeruginosa* (0.7% in 2003–2004 to 1.8% in 2011–2012; *p =* 0.044), and *Morganella morganii* (0% in 2003–2004 to 0.8% in 2011–2012; *p =* 0.025) [10].

When compared with patients infected with other organism(s), patients infected with Pseudomonas, MRSA, and Proteus had significantly decreased infection-free rates. Infection with methicillin-sensitive *Staphylococcus aureus* (MSSA), coagulase-negative Staphylococcus, MRSA, Pseudomonas, Peptostreptococcus, Klebsiella, Candida, Diphtheroids, *Propionibacterium acnes*, and *Proteus* spp. was associated with 1.13–2.58 additional surgeries, whereas MSSA, coagulase-negative Staphylococcus, Proteus, MRSA, Enterococcus, Pseudomonas, Klebsiella, beta-hemolytic Streptococcus, and Diphtheroids were associated with 8.56–24.54 additional days in hospital for infection. Although fewer, Gram-negatives had a greater risk for complications even compared to MRSA; patients with *Pseudomonas* and *Proteus* were at greater risk for extensive hospitalization, multiple surgical procedures, and treatment failure of hip PJI [11].

Although no significant association between antibiotic resistance and biofilm formation was shown, MDR and XDR isolates were found to be greater biofilm formers than non-MDR isolates and their ability to form biofilm differed among species, with *Pseudomonas aeruginosa* being the strongest biofilm producer [12]. XDR GNB and comorbidities were independently associated with MDR GNB PJI treatment failure [13]. The predisposing factors for MDR GNB-associated PJIs include revision arthroplasties, previous orthopedic infections, postoperative hematomas, and early infections.

In the largest multicenter international cohort (2000–2015) of MDR/XDR GNB, 131 patients with PJI were included. The patients were elderly (mean age 73), with comorbidities (58.8%). MDR (*n* = 108) and XDR (*n* = 23) were assessed. The most common pathogens were *Escherichia coli* (33.6%), *Pseudomonas aeruginosa* (25.2%), *Klebsiella pneumoniae* (21.4%), and *Enterobacter cloacae* (17.6%). *Pseudomonas aeruginosa* predominated in XDR cases (50%). Isolates were identified as carbapenem resistant (*n* = 12), fluoroquinolone resistant (*n* = 63), and ESBL producers (*n* = 94) [14]. Moreover, 57 cases of osteosynthesis-associated infection (OAI) by MDR/XDR GNB were revealed. Those patients had a history of trauma (87.7%) or tumor resection (7%). Pathogens included *Escherichia coli* ESBL producer (*n* = 16), *Pseudomonas aeruginosa* (*n* = 14), *Enterobacter* spp. (*n* = 9), *Acinetobacter* spp. (*n* = 5), *Klebsiella* spp. (*n* = 7), *Proteus mirabilis* (*n* = 3), *Serratia marcescens* (*n* = 2), and *Stenotrophomonas maltophilia* (*n* = 1). XDR accounted for 50% and 40% of *Pseudomonas aeruginosa* and *Acinetobacter* spp. strains, respectively [15]. A very recent multicenter national study of 44 patients with BJI by fluoroquinolone-resistant GNB revealed that Enterobacteriaceae were responsible for 61% and *Pseudomonas* spp. for 39% of cases, with an overall rate of MDR/XDR GNB infections of 61% [16].

Resistant GNB as causative pathogens in bone and joint infections have also been reported among young children (under 4 years old), underscoring the importance of careful consideration when choosing empirical treatment [17]. Epidemiological surveillance of multidrug resistance in BJIs is mandatory both for Gram-positive and Gram-negative bacteria.

## 3. Potent Antibiotics in BJIs by MDR/XDR

BJIs with or without an orthopedic device are characterized by the formation of biofilm. As genetically defined pathogens, both GPB and GNB are able to induce biofilm formation on the surface of the implant, but also intraosseously. A variety of in vitro and in vivo models have been developed to better identify and depict the role of biofilm in BJIs [18,19]. The low concentration of oxygen and nutrients in the biofilm environment leads to heterogeneous phenotypic changes in the bacteria, with antimicrobial tolerance being of paramount importance [20]. In this narrative review, we will shortly present antimicrobials designated to combat MDR and XDR GNB in patients with BJIs. Β-lactams demonstrated reduced concentrations in bone compared to plasma whilst fluoroquinolones sufficiently penetrate to bone tissue [21]. Table 1 summarized the mode of action and antimicrobial spectrum along with the limitations of use of currently available antibiotics against MDR/XDR GNB in BJIs (Table 1).

### 3.1. Carbapenems

Carbapenems, a subclass of β-lactam antibiotics, act as inhibitors of the peptidase domain of penicillin-binding proteins (PBPs) and are resistant to hydrolysis by β-lactamases [22]. They possess the broadest spectrum of activity among β-lactam antibiotics and are the antibiotics of choice for treating ESBL [23]. However, it is noteworthy that ertapenem is inactive against *Pseudomonas aeruginosa* in contrast to meropenem and imipenem–cilastatin, which demonstrate antipseudomonal efficacy [22].

In the context of BJIs, carbapenems exhibit promising pharmacokinetic properties. Pharmacokinetic studies evaluating the penetration of antibiotics into bone and joint tissues indicated that carbapenems achieve good penetration into these compartments [17,21]. Moreover, an in vitro study showed that the combination of the continuous infusion of meropenem with colistin maximized the anti-biofilm effect [24]. Similarly, in an in vivo Pk/Pd model, imipenem and colistin showed anti-biofilm activity [25]. However, in experimental osteomyelitis, meropenem was not effective in carbapenem-resistant strains of *Klebsiella pneumoniae*, even when co-administered with colistin [26].

### 3.2. Meropenem–Vaborbactam

Meropenem–vaborbactam is a combination that involves a broad spectrum carbapenem and a novel β-lactamase inhibitor. Approved in 2017 for the treatment of complicated urinary tract infections, it demonstrated non-inferiority compared to piperacillin–tazobactam. Vaborbactam is a novel β-lactamase inhibitor that protects meropenem against carbapenemases such as KPC. However, this combination is ineffective against metallo-β-lactamases (MBLs) or OXA-type carbapenemases. Additionally, it does not provide activity against *Pseudomonas aeruginosa* and *Acinetobacter baumannii* strains already resistant to meropenem [27].

### 3.3. Imipenem–Cilastatin–Relebactam

Imipenem–cilastatin–relebactam is a combination of a broad spectrum carbapenem and a novel β-lactamase inhibitor. Approved in 2019, relebactam enhances the activity of imipenem–cilastatin against AmpC and KPC-producing Enterobacteriaceae as well as *Pseudomonas aeruginosa* [27]. However, this combination is not effective against MBL [27]. Relebactam may provide a modest enhancement to the activity of imipenem against OXA-type CRE [28]. It does not provide activity against *Acinetobacter baumannii* strains already resistant to imipenem–cilastatin [27].

Although specific data on the penetration of β-lactamase inhibitors into bones and joints are limited, these inhibitors are believed to share similar pharmacokinetic profiles with the β-lactam antibiotics they accompany [29].

### 3.4. Ceftolozane–Tazobactam

Ceftolozane–tazobactam is a relatively recent combination of a fifth-generation cephalosporin and a β-lactamase inhibitor, approved in 2014. Ceftolozane is a new cephalosporin and its combination with tazobactam broadens its activity against ESBL-E and MDR *Pseudomonas aeruginosa* [27]. It is approved for treating adults with complicated intra-abdominal infections, complicated urinary tract infections including pyelonephritis, hospital acquired bacterial pneumonia, and ventilator-associated bacterial pneumonia [27].

Post-marketing studies, however, have suggested efficacy against bone and joint infections. In an in vitro pharmacodynamic biofilm model, monotherapy with ceftolozane–tazobactam showed limited anti-biofilm activity against susceptible MDR *Pseudomonas aeruginosa* strains, raising considerations about its effectiveness against *Pseudomonas aeruginosa* biofilm [30]. The combination of ceftolozane–tazobactam with colistin showed enhanced activity in an in vitro pharmacodynamic biofilm model with a meropenem-resistant *Pseudomonas aeruginosa* strain [30].

### 3.5. Ceftazidime–Avibactam

Ceftazidime–avibactam is a relatively recent combination of a third-generation cephalosporin plus β-lactamase inhibitor, approved in 2015. It is indicated to treat adults with complicated intra-abdominal infections, complicated urinary tract infections including pyelonephritis, hospital-acquired bacterial pneumonia, and ventilator-associated bacterial pneumonia. Ceftazidime is effective against GNB, including Pseudomonas aeruginosa. The combination of ceftazidime–avibactam has shown activity against some DTR P. aeruginosa isolates but the resistance rates exceed 50% in Acinetobacter baumannii [28]. When combined with the novel β-lactamase inhibitor, avibactam, the drug regains activity against several β-lactamases, such as ESBL-E and KPC, as well as OXA-48, but it lacks activity against MBL [27].

In an experimental rabbit model of osteomyelitis caused by OXA-48-/ESBL-producing Escherichia coli, ceftazidime–avibactam (CAZ-AVI) significantly reduced bacterial counts compared to controls such as monotherapy, although no difference in bone sterilization was observed. However, when CAZ-AVI was used in combination with colistin (91%) or fosfomycin (100%) or gentamicin (100%), bone sterilization was achieved [31]. In another rabbit model of osteomyelitis by the Carbapenemase-producing Klebsiella pneumonia (CPKP) strain, the best efficacy in the eradication of infection was demonstrated by the combination of CAZ-AVI with gentamicin [32].

### 3.6. Cefiderocol

Cefiderocol is an injectable siderophore cephalosporin, approved in 2019, with a unique mechanism that enables it to be actively transported across the outer membrane of Gram-negative bacteria via ferric iron transport systems [33]. Its siderophore-like property also provides enhanced stability against β-lactamases. Cefiderocol demonstrates broad coverage of GNB, including *Acinetobacter baumannii*, *Pseudomonas aeruginosa*, *Stenotrophomonas maltophilia*, and resistant Enterobacterales, with activity against all four Ambler class β-lactamases (A, B, C, and D) [34]. It has been approved for treating infections caused by aerobic GNB in adults with limited treatment options.

In an in vitro study, the anti-biofilm activity of cefiderocol alone and in combination with imipenem against *Pseudomonas aeruginosa* strains was evaluated. Cefiderocol and imipenem alone showed poor anti-biofilm activity, but their combination exhibited enhanced anti-biofilm activity [35]. A pharmacokinetic study of XDR *Pseudomonas aeruginosa* bacteremia, and presumed osteomyelitis treated with cefiderocol, demonstrated that cefiderocol concentrations in bone and skin/subcutaneous tissue were sufficient for adequate drug penetration in these compartments, and the patient fully recovered [36].

### 3.7. Fosfomycin

Fosfomycin is a broad spectrum antibiotic that inhibits bacterial cell wall synthesis by targeting the enzyme UDP-N-Acetylglucosamine Enolpyruvyl Transferase (MurA). It is effective against both GPB and GNB, including MDR and XDR Enterobacteriaceae and *Pseudomonas aeruginosa* [37]. However, its use in monotherapy is generally not recommended due to the rapid emergence of resistance and its reduced efficacy in the presence of a high bacterial inoculum, a characteristic of some bone and joint infections [37,38].

Experimental studies of bone and joint infections have demonstrated that fosfomycin monotherapy is more effective than tigecycline, gentamicin, and colistin monotherapy against ESBL-producing *Escherichia coli*. Nevertheless, its combination with other antibiotics has yielded superior results. In vitro findings highlight the promising potential of fosfomycin in combination with carbapenems for treating ESBL/CRE and *Pseudomonas aeruginosa* [38]. The in vitro synergism of fosfomycin with colistin significantly reduced bacterial biofilm by *Klebsiella pneumoniae* and *Pseudomonas aeruginosa* [39].

### 3.8. Colistin

Polymyxins consist of polymyxins A–E, with polymyxin B and polymyxin E (colistin) being the two commercially available forms [40]. Colistin acts by binding lipopolysaccharides (LPSs) and phospholipids in the outer membrane of GNB, disrupting the membrane, leading to the leakage of intracellular contents, and causing bacterial death [40]. Its spectrum of activity is against aerobic GNB, including MDR pathogens such as Enterobacteriaceae, *Pseudomonas aeruginosa,* and *Acinetobacter baumannii* [41]. For invasive infections caused by CRE, colistin is recommended to be used in combination with one or more additional agents to which the pathogen displays a susceptible MIC [41]. Colistin is active in fewer metabolic cells in the deeper layer of the biofilm structure. High doses might be necessary in order to combat the hetero-resistance of Pseudomonas and Enterobacteriaceae. The combination with other potent antibiotics (e.g., fosfomycin, β-lactam, aminoglycoside) increased success rates (up to 80%) both experimentally and clinically [20]. An experimental model of osteomyelitis due to KPC demonstrated treatment failure with colistin monotherapy in both in vivo and in vitro studies but effectiveness with combinations [42]. All relevant experimental data support the combination of colistin with various antimicrobials (i.e., carbapenems, CAZ-AVI, ceftolozan–tazobactam, fosfomycin) [24,25,30,32,39].

### 3.9. Tigecycline

Tigecycline is a bacteriostatic antibiotic that inhibits protein synthesis and belongs to the tetracycline class. It exhibits board spectrum activity against GPB and GNB, including MDR Enterobacterales and *Acinetobacter baumannii*. However, it is not effective against *Pseudomonas aeruginosa* [43]. Tigecycline is approved by the US Food and Drug Administration (FDA) and by the European Medicines Agency (EMA) for the treatment of complicated intra-abdominal infections and complicated skin and soft tissue infections, and in the case of the FDA, community-acquired pneumonia [44]. An in vivo experimental model of foreign body infection comparing tigecycline, colistin, and fosfomycin revealed that combinations of those agents maximized the eradication of infection without bacterial regrowth [45].

The scarcity of experimental and pre-clinical data on the effectiveness of older and novel antimicrobials in BJIs raises considerations regarding their true potency in difficult to treat MDR GNB. Most BJIs are caused by Gram-positive bacteria with more available research data.

## 4. Clinical Data on Antimicrobial Treatment of BJIs by MDR GNB

Clinical data from large epidemiological studies exclusively on MDR/XDR GNB bone and joint infections with and without orthopedic implants are scarce. Moreover, randomized controlled studies comparing antibiotics for those infections are lacking. Therefore, our experience is based on some cohort studies or case series or well-documented case reports. Overall, the treatment of BJIs by MDR/XDR GNB is based on experts’ opinion and the off-label use of all available susceptible antibiotics.

Regarding antimicrobial treatment, multiple schemes have been proposed based on pathogens’ susceptibility and antibiotics’ availability. Treatment was mainly given intravenously for prolonged periods of time. Treatment was individualized and highly varied among studies. In the past, a combination of ceftazidime (3 g/day) plus ciprofloxacin (1.5 g/day) for 6 weeks followed by ciprofloxacin (1.5 g/day orally) for 6 months was an adequate successful treatment for *Pseudomonas aeruginosa* PJI and other orthopedic device-related infection. However, only 14 patients were eligible for analysis [46]. Another study including 24 patients reported a good outcome with the combination of cefepime plus fluoroquinolone for pseudomonal BJI [47]. Fluoroquinolones were evaluated as a prompt treatment—both intravenously and orally—for BJI by GNB because of their good bioavailability and sufficient penetration to the bone tissue. Osteomyelitis by *Enterobacter cloacae* producing ESBL and Amp-C was successfully treated by ciprofloxacin with or without surgical debridement [48]. Resistance to fluoroquinolones emerges as a major restriction in the treatment of BJIs by GNB; fluoroquinolones were independently associated with treatment success in a large cohort of 242 patients with PJIs caused by GNB (19% resistant to ciprofloxacin). Ciprofloxacin treatment exhibited an independent protective effect (adjusted hazard ratio (aHR) 0.23; 95% CI, 0.13–0.40; *p* < 0.001) with 79% success even in patients treated with DAIR [49]. This result was corroborated by another study including 34 patients with GNB PJIs treated with DAIR [50]. Therefore, there is enough evidence supporting the use of fluoroquinoline as a cornerstone of antimicrobial treatment in BJIs by GNB. However, another study comprising 76 surgically treated patients with PJI by GNB did not demonstrate differences in the outcome with or without fluoroquinolones. The low failure rate (21%) observed in patients not receiving fluoroquinolones was attributed to the standardized attitude of maintaining intravenous β-lactams throughout treatment duration (median = 90 days) [51]. Therefore, it is questionable if the continuous infusion of β-lactams can counterpoise the lack of fluoroquinolones from the therapeutic quiver.

In a series of 34 patients (68% by XDR *Pseudomonas aeruginosa*), the continuous use of b-lactams in combination with colistin was more effective than monotherapy; however, overall success was attained only in 50% of cases [52]. Of notice, colistin methanesulphonate was administered at a total daily dose of 6 million units and the dose of the antipseudomonal b-lactam was chosen upon the lowest MIC in order to achieve a target drug concentration at and above the MICs of the pathogen [52]. This is a well-documented clinical study on the benefits of combination treatment over monotherapy for XDR *Pseudomonas aeruginosa* BJI. Few data support that antimicrobial combination seems to be effective even in cases of implant retention following debridement [10]. In a cohort of 44 patients with BJI by GNB resistant to fluoroquinolones, colistin plus intravenous beta-lactam were introduced for a median of 28 days, followed by intravenous beta-lactam alone for 19 days (IQR 5–35). The overall cure rate in a 24-month follow-up was 82% (95% CI 68–90%) and 80% (95% CI 55–93%) in patients with implant retention [16]. However, the prolonged use of colistin raised concerns for its toxicity. In a multicenter cohort of 19 patients treated for BJI by MDR/XDR GNB, the prolonged administration of colistin (median 81 days) led to a rate of 73% of successful outcome whilst renal insufficiency was restored in all patients after the end of treatment [53]. The in vitro synergy of colistin with other antibiotics had successfully guided antimicrobial treatment in a patient with polymicrobial XDR GNB PJI [54]. However, the emergence of resistance to the drug has been described under treatment for KPC BJI [55]. In the largest multicenter international cohort of patients with PJIs by MDR/XDR GNB, colistin methanesulphonate was administered mostly in patients with XDR PJI (69.5%) compared with MDR PJI (11.1%) at a daily dose that ranged from 2 × 10^6^ IU to 9 × 10^6^ IU adjusted to renal function [14]. Treatment success was more evident in MDR (66.7%) than XDR (39.1%) cases despite the use of colistin in the last ones (*p* = 0.018). Neither the total length of treatment (median 74.3 days) nor the use of combined antibiotics has influenced the outcome of patients. DAIR was independently related with treatment failure in that study. Non-DAIR procedures demonstrated an independent favorable impact on successful outcomes (OR = 0.23, 95% CI 0.10–0.53; *p* = 0.001). The superiority of non-DAIR surgical procedures remained unchangeable by the time of infection onset (early/late), type of resistance (MDR/XDR), and antimicrobial treatment (colistin versus non-colistin) [14]. Despite limitations, this large study confers to the current knowledge on the utility of colistin in those DTR infections. In accordance, a smaller study of 25 patients with either PJI or osteosynthesis-related infection demonstrated that implant retention was successful only in 33% vs. 100% in cases of implant removal [56]. Treating Gram-negative PJIs with debridement was associated with a lower 2-year cumulative probability of success than treating Gram-positive PJIs with debridement (27% vs. 47%; *p* = 0.002) [57]. In a large multicenter study including only patients with osteosynthesis-associated infections (OAIs), the type of antimicrobials and the type of resistance (MDR vs. XDR) did not affect the outcome of patients. Meropenem was given in the case of MDR GNB susceptible to carbapenems and colistin in the case of XDR GNB. Only an age > 60 years (HR of 3.875; 95% CI of 1.540–9.752; *p* = 0.004) and multiple surgeries for OAI (HR of 2.822; 95% CI of 1.144–6.963; *p* = 0.024) were associated with treatment failure [15]. The main advantage of this study is the focus only on long bone osteosynthesis infection instead of mixing up patients with PJI and fracture-related infections (FRIs) as they largely differ in terms of pathogenicity and outcome.

Tigecycline alone or in combination with colistin or aminoglycoside was administered in two cases of osteomyelitis caused by KPC. The survival rate was low due to multidisciplinary health complications [58]. Oliveira et al. conducted a retrospective study comparing tigecycline and colistin as monotherapy in patients with carbapenem-resistant *Acinetobacter baumannii* complex osteomyelitis. Tigecycline demonstrated a better safety profile than colistin, with no significant difference in clinical outcomes at a 12-month follow-up. Favorable outcomes were observed in 38.7% of tigecycline-treated patients compared to 44.1% of those treated with colistin [59]. Vila et al. also documented three cases of *Acinetobacter baumannii* prosthetic joint infections successfully treated with high maintenance doses of tigecycline, achieving favorable outcomes [60]. However, relevant data are fragmentary, based only on small case series.

Fosfomycin is a portent antibiotic both against GNB and GPB. Recently, it has proven to be effective in a large cohort of patients (including 17% of patients with BJI) with DTR MDR *Pseudomonas aeruginosa* and *Klebsiella* spp. infections. Fosfomycin was susceptible even in the presence of resistance of the novel b-lactams (i.e., CAZ-AVI and ceftolozane–tazobactam). A combination of fosfomycin with any other potent antibiotic led to 100% treatment success in patients with BJIs [61]. A patient with MDR *Pseudomonas aeruginosa* osteomyelitis was successfully treated with the combination of intravenous fosfomycin and ceftolozane–tazobactam followed by meropenem [62]. Treatment was successful in limb-threatening osteomyelitis due to XDR *Pseudomonas aeruginosa* with aggressive surgical debridement and a combination of intravenous fosfomycin and colistin [63]. Despite current knowledge regarding the efficacy of fosfomycin in MDR GNB, the data usually comprise mixed-up patients with various infections whilst solely BJIs by MDR/XDR GNB include case reports.

Data on the use of novel β-lactams for BJIs by MDR GNB are provided only by sporadic case reports and small case series. Seven patients with osteomyelitis were included in the EZTEAM real-world study, which evaluated the administration of CAZ-AVI in 516 patients [64]. CAZ-AVI was successfully given for 6 weeks in combination with either colistin, fosfomycin, or amikacin in three patients with complicated osteomyelitis [65]. In a multicenter retrospective study, 9.8% (4 out of 41) of patients with BJIs by GNB—mainly *Pseudomonas aeruginosa—*treated with CAZ-AVI presented with a 100% clinical cure rate [66]. CAZ-AVI was also successfully given for various time intervals in patients with knee PJIs and without significant, non-reversible adverse events [67]. Two cases of orthopedic implant infection by KPC and MBL producers were treated with a combination of aztreonam in a continuous infusion of 3 g per 12 h, twice a day for 12 weeks, CAZ-AVI in a continuous infusion of 3 g/0.75 g per 12 h, twice a day for 12 weeks, and fosfomycin 3 g/8 h at a discontinuous infusion for the first 4 weeks post-surgically [68]. Additional case reports describe the bone and joint infections caused by NDM-producing *Klebsiella pneumonia* [69] and MBL-producing *Pseudomonas aeruginosa* [70], both successfully treated with the combination of CAZ-AVI and aztreonam and the combination of CAZ-AVI, aztreonam, and amikacin, respectively. CAZ-AVI for 6 weeks was effective in cases of vertebral osteomyelitis by XDR GNB [71,72,73].

Another novel b-lactam, ceftolozane–tazobactam (C/T) was effective in patients with BJI by XDR *Pseudomonas aeruginosa* BJI as outpatient parenteral antimicrobial treatment (OPAT) via continuous infusion through elastomeric pumps [74]. In an observational study, Rempenault et al. reported a 60% successful treatment of bone and joint infections (3 out of 5 patients). All cases involved at least one MDR *Pseudomonas aeruginosa* infection and 60% were polymicrobial [75]. In a multicenter retrospective study, patients who received C/T as OPAΤ were assessed. Among them, 27% (n = 34) had BJIs, treated successfully in 72.7% of cases [76].

In a retrospective case series analyzing the use of meropenem–vaborbactam for CRE infections, five out of fifteen patients (33.3%) were treated for BJIs. Among those with CRE-related BJIs, three patients (60.0%) showed a positive clinical outcome [77]. Few BJI cases were included in another real-world study with CRE infections treated by meropenem–vaborbactam [78]. Clinical data on meropenem–vaborbactam in BJIs are scarce.

Imipenem–cilastatin–relebactam showed favorable outcomes in patients with various infections, including bone infections, caused by KPC-producing *Klebsiella pneumoniae* and DTR *Pseudomonas aeruginosa*, treated with imipenem–cilastatin–relebatatm [79]. However, relevant data for BJIs are lacking. Recently, the first patient with vertebral spondylitis by XDR Enterobactercloacae was successfully treated with the prolonged infusion of imipenem–cilastatin–relebactam, followed by meropenem–varbobactam [29]. Another novel b-lactam, cefiderocol, was successfully initiated as a last-resort treatment in two patients with BJI by XDR *Acinetobacter* spp. [80,81]. The 8-week combination of cefiderocol with trimethoprim–sulfamethoxazole (TMP/SMX) was proven to be successful in the treatment of knee PJIs complicated by MDR *Stenotrophomonas maltophilia* [82]. In a case of osteomyelitis caused by NDM-producing *Pseudomonas aeruginosa*, cefiderocol was given as OPAT treatment with success [83]. The long-term compassionate use of cefiderocol for 14 weeks was proven safe and effective in an infant with BJI and an adult with cranial osteomyelitis by XDR *Pseudomonas aeruginosa* [84,85]. In a multicenter study, cefiderocol demonstrated similar efficacy to the best available treatment for infections caused by carbapenem-resistant GNB. However, a higher number of deaths were reported in the cefiderocol group, particularly among patients with *Acinetobacter* spp. infections [86].

Overall, robust relevant data of any of the novel b-lactams against MDR/XDR GNB in BJIs do not exist for the time being.

Similarly, data are very scarce regarding local antibiotic elution in cases of BJIs by MDR/XDR GNB. Two patients were successfully treated by colistin-impregnated cement along with intravenous colistin [87,88]. Another patient who underwent two-stage revision for a hip PJI caused by MDR *Serratia marcescens* was treated with systemic meropenem along with local elution of the impregnated drug [89]. Colistin-impregnated cement along with intravenous administration of the drug was successfully given in a patient with CRAB (Carbapenem-resistant *Acinetobacter baumannii*) [90]. A case of XDR GNB BJI treated by impregnated cement with cefiderocol along with intravenous administration was proven to be effective [82]. Promising results from a pre-adapted bacteriophage with meropenem and colistin, followed by CAZ-AVI administration, were reported in a 30 year old bombing victim with a fracture-related PDR *Klebsiella pneumonia* infection [91].

Overall, novel antibacterials can be a partner in DTR GNB in the absence of other susceptible compounds or in combination with older antibiotics such as colistin, tigecycline, and fosfomycin. Although very limited, current data support the combination treatment over monotherapy.

## 5. Discussion

GNB along with antimicrobial resistance are emerging worldwide [1,10]. Few antimicrobial options are available in our armamentarium against MDR/XDR GNB BJI. Most cases are due to *Escherichia coli*, ESBL producers, and DTR *Pseudomonas aeruginosa*. However, the global emergence of carbapenem-resistant *Klebsiella pneumoniae*, *Enterobacter* spp., and *Acinetobacter* spp. raises important concerns about antimicrobial therapeutic options. Pre-clinical and clinical data on BJI by MDR/XDR GNB are relatively scarce, mostly based on observational retrospective or prospectively performed analyses [92]. There is a lack of randomized controlled trials directly comparing different treatment regimens and treatment duration. Radical debridement and implant removal are the optimal therapy options in patients with orthopedic implant infection by DTR pathogens. This option is supported by the largest multicenter studies on PJI and osteosynthesis-related infections by MDR/XDR GNB [14,15]. For DTR GNB, we summarized the current opinion on the use of last-resort antibiotics, including older ones (i.e., colistin, tigecycline, fosfomycin) and newer ones (i.e., ceftazidmie–avibactam, ceftolozane–tazobactam, meropenem–varobactam, imipenem–cilastatin–relebactam, and cefiderocol). In an attempt to optimize antimicrobial treatment, the use of the continuous infusion of β-lactams in combination with colistin or other potent drugs was suggested and seems to be efficacious. However, therapeutic drug monitoring is mandatory in order to achieve maximum doses for the maximum time above the pathogens’ MIC values [16,51,52]. Older antibiotics such as fosfomycin, tigecycline, and colistin along with newer ones such as novel β-lactam/β-lactamase inhibitors are an effective option in MDR and DTR GNB [93]. Fosfomycin is a safe and promising agent in combating both GPB and GNB in bone and joint infections, however only in combination with other potent antibiotics whilst the appropriate dosing is not established [94]. Colistin is a significant compound in XDR GNB BJI for susceptibility reasons, bactericidal properties, and biofilm combating results. However, the risk of adverse events often leading to the discontinuation of the drug is not negligible [20]. Few data exist for tigecycline but if susceptible, it seems that it is effective for DTR BJI at the highest possible dose, in particular in combination with other antibiotics; however, it has intrinsic resistance for *Pseudomonas aeruginosa*. Regarding novel β-lactam and β-lactam/β-lactamase inhibitors, relevant data are scarce based on case reports, small case series, and very few experimental data. However, these antibiotics can be used as a last resort of treatment in cases of desperate XDR BJI always after prompt surgical treatment. We would recommend that novel antimicrobials would be better being introduced in combination with older potent drugs (colistin, fosfomycin, or tigecycline) based on a susceptibility test, the co-existence of GPB, and the drug safety profile adjusted to each patient. Compliance with longer than usual treatment duration and continuous clinical and laboratory monitoring for drugs’ adverse events is mandatory both for older and newer antibiotics. The preparation of homemade local drug elution systems for DTR GNB should be based on the collaboration of surgeons with clinical microbiologists and infectious disease specialists. Local treatment should be always combined with intravenous antibiotic administration. More clinical and laboratory research should be provided in the field in order to ensure optimized drug delivery into bones and joints and increase the rates of clinical cure. Large multicenter prospective clinical trials are mandatory. For the time being, in general, the antimicrobial treatment for DTR GNB in BJIs could follow the published guidelines for all MDR/XDR GNB [95].

## Figures and Tables

**Table 1 pathogens-14-00130-t001:** Highlights of antibiotics for multidrug-resistant (MDR) Gram-negative bacteria (GNB).

Antibiotic	Mechanism of Action	Spectrum of Activity of Antibiotics for MDR GNB	Limitations
Carbapenems (e.g., meropenem, imipenem–cilastatin, ertapenem)	Inhibit bacterial cell wall synthesis by binding to PBPs	ESBL	No activity against carbapenemases, DTR *A. baumannii*, DTR *P. aeruginosa* Ertapenem is inactive against *P. aeruginosa*
Meropenem– vaborbactam	Inhibits bacterial cell wall synthesis; vaborbactam inhibits KPC-producing β-lactamases	ESBL ^1^, KPC	No activity against MBL- or OXA-type carbapenemases, DTR *P. aeruginosa,* DTR *A. baumannii*
Imipenem– cilastatin– relebactam	Inhibits bacterial cell wall synthesis; relebactam inhibits KPC	ESBL ^1^, KPC, Relebactam may slightly enhance the activity of imipenem against OXA–carbapenemases, DTR *P. aeruginosa*	No activity against MBL
Ceftolozane– tazobactam	Inhibits bacterial cell wall synthesis; tazobactam inhibits β-lactamases	ESBL, DTR *P. aeruginosa*	No activity against carbapenemases, DTR *A. baumannii,* AmpC β-lactamases
Ceftazidime– avibactam	Inhibits bacterial cell wall synthesis; avibactam inhibits β-lactamases, including KPC and OXA-48 carbapenemase	ESBL, KPC, AmpC β-lactamases, OXA-48 carbapenemase, DTR *P. aeruginosa*	No activity against MBL High resistance rates in *A. baumannii*
Cefiderocol	Siderophore cephalosporin: actively transported into bacteria via iron transport systems	ESBL, KPC, MBL, AmpC β-lactamases, OXA-48 carbapenemase, DTR *P. aeruginosa*, DTR *A. baumannii*	
Fosfomycin	Inhibits bacterial cell wall synthesis by targeting MurA enzyme	ESBL, CRE (all classes of carbapenemases, including MBL), DTR *P. aeruginosa*	Its use as monotherapy is generally not recommended
Colistin	Disrupts bacterial cell membrane integrity by binding to LPS and phospholipids in the outer membrane of GNB bacteria	ESBL, CRE (all classes of carbapenemases, including MBL), DTR *P. aeruginosa,* DTR *A. baumannii*	It is recommended to be used in combination with one or more additional agents to which the pathogen displays a susceptible MIC
Tigecycline	Inhibits protein synthesis by binding to the bacterial 30S ribosomal subunit	ESBL, CRE (all classes of carbapenemases, including MBL), DTR *A. baumannii*	It is not effective against *P. aeruginosa*

PBPs: penicillin-binding proteins; DTR: difficult to treat; KPCs: Klebsiella pneumoniae carbapenemases; ESBLs: extended-spectrum beta-lactamases; MBLs: metallo-β-lactamases; CRE: carbapenem-resistant Enterobacteriaceae; LPSs: lipopolysaccharides; GNB: Gram-negative bacteria; MIC: minimal inhibitory concentration. ^1^ The established therapeutic option for severe infections due to ESBL is a carbapenem without a β-lactamase inhibitor.

## Data Availability

Data are contained within the article.

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
