# Peer review of "Antimicrobial Treatment Options for Multidrug Resistant Gram-Negative Pathogens in Bone and Joint Infections"

_pathogens, 2025, doi:10.3390/pathogens14020130_

Round 1

Reviewer 1 Report

Comments and Suggestions for Authors

It is a well written manuscript with a lot of useful information.

Discussion section is too short. The existing literature is enough to support the text improvement. 

Language editing

Author Response

Comment 1. It is a well written manuscript with a lot of useful information.

Response 1. Thank you for validating our manuscript

Comment 2.Discussion section is too short. The existing literature is enough to support the text improvement. 

Response 2: Thank you for your comment. Discussion was elongated adding also  experts’ opinion according to your suggestion

Reviewer 2 Report

Comments and Suggestions for Authors

The authors have made a good attempt in compiling relevant literature on the antimicrobial treatment including pathogens involved in bone and joint infections. However, the following points are for authors' considerations:

1. It is worth including recent estimates on the prevalence and characteristics of the multi-drug resistant genes involved in causing drug resistance.

2. How do you ensure that you have comprehensively included all the pertinent articles relevant to this topic? It would provide more trust to the readers if you can use appropriate search strategy including the databases used for obtaining the articles used in your manuscript.

3. Specify the key research questions that you are addressing in this manuscript.

4. Critically examine and comment on the quality of studies included in your manuscript.

5. At the end of every sub-topic, provide the key highlights that includes the strength, weakness and key recommendations related to that topic.

6. The antimicrobials are just listed with one or two studies being cited for their use in BJI. A more comprehensive literature review encompassing all key studies carried out with each antimicrobial drug is needed.

7. Provide recommendations for clinicians and researchers exploring this topic.

Author Response

The authors have made a good attempt in compiling relevant literature on the antimicrobial treatment including pathogens involved in bone and joint infections. However, the following points are for authors' considerations:

Comment 1. It is worth including recent estimates on the prevalence and characteristics of the multi-drug resistant genes involved in causing drug resistance.  

Response 1. Thank you for your  comment. We added at the introduction the global estimates of the incidence of MDR GNB based also on  resistance mechanisms. Detailed analysis of genes escapes from the aim of the present study, however, we included relevant information into the text (section 3.) and in a table comprising the potency of older and novel antibiotics   against MDR GNB according to the genetic type of resistance

Comment 2. How do you ensure that you have comprehensively included all the pertinent articles relevant to this topic? It would provide more trust to the readers if you can use appropriate search strategy including the databases used for obtaining the articles used in your manuscript.

Response 2. We thank  a lot the reviewer for this comment. We performed a thorough search by key words in pubMed : “ bone and joint infections” AND “ multidrug resistant” AND “ Gram negative bacteria” , terms. Besides, our search was specified  by type of bone and joint infection  ( “ osteomyelitis”, “ septic arthritis”, “prosthetic joint infections”, “ fracture related infections”, “ osteosynthesis” ) by bacterial species and resistance  mechanism (e.g” Pseudomonas aeruginosa”, “Klebsiela KPC”, “carbapenem resistant Acinetobacter”, “Esherichia coli”, “ Enterobacter spp” “ ESBL”, “fluoroquinolone resistant”) and by  type of resistance (“ multidrug resistant”, “extensively drug resistant”, “difficult to treat” ). Furthemore, some additional  data was also  retrieved  by the references of papers relevant to experimental, epidemiological and clinical data.  The text was amended accordingly in  the Introduction section.

Comment 3. Specify the key research questions that you are addressing in this manuscript.

Response 3. Please see  reponse in comment 2

Comment 4. Critically examine and comment on the quality of studies included in your manuscript.

Response 4. We thank the reviewer for this comment. Although we have already included the  scarcity of large trials as a limitation, we tried furthermore to separately comment on the most  important studies based on the size of the study, the rarity of cases and the input on current knowledge. Those changes are made in the revised text.

Comment 5. At  the end of every sub-topic, provide the key highlights that includes the strength, weakness and key recommendations related to that topic.

Response 5. We  thank a lot the reviewer for this comment.  We added this statement at the end of each sub-topic. Changes are included  into the revised text.

Comment 6. The antimicrobials are just listed with one or two studies being cited for their use in BJI. A more comprehensive literature review encompassing all key studies carried out with each antimicrobial drug is needed. 

Respone 6. We are thankful for this comment. By revising section 3 we added more information about novel antimicrobials by including other recent experimental studies for BJI by GNB. Relevant references were also included.

Comment 7. Provide recommendations for clinicians and researchers exploring this topic.

Response 7. Thank you for your comment. Recommendations, based on current data and personal expertise on the field are provided at the discussion with reserve due to the scarcity of published data. Please find amended text in the discussion section.

Thank you for  all your fruitful comments which helped us to significantly improve our manuscript

Reviewer 3 Report

Comments and Suggestions for Authors

Thank you for your invitation to review this manuscript.

This review summarizes the epidemiology and treatment options of multidrug resistant (MDR) and extensively drug resistant (XDR) Gram-negative bacteria (GNB) in bone and joint infections (BJI) in a narrative review.

Recommendation:

1.     It is suggested to add one or two summary tables to compare the mechanism of action, indications, advantages and limitations of different antibacterial drugs, so as to improve the intuitiveness and readability of the article.

2.     It is suggested that more specific summaries of research data be included in the abstract to enhance readers' interest in the full text.

Author Response

This review summarizes the epidemiology and treatment options of multidrug resistant (MDR) and extensively drug resistant (XDR) Gram-negative bacteria (GNB) in bone and joint infections (BJI) in a narrative review.

Recommendation:

Comment 1.     It is suggested to add one or two summary tables to compare the mechanism of action, indications, advantages and limitations of different antibacterial drugs, so as to improve the intuitiveness and readability of the article.  

Response 1. We thank the reviewer for this suggestion. In the revised manuscript, we provide a  table summarizing main mode of actions, susceptibility patterns and restrictions according to resistance mechanisms for all antimicrobials which are reported  in the text (table 1).

Comment 2.     It is suggested that more specific summaries of research data be included in the abstract to enhance readers' interest in the full text.

Respoonse 2. We thank the reviewer for this comment. Abstract was amended accordingly in order to include main points of our review regarding  antimicrobials for MDR/XDR GNB in BJI.

Round 2

Reviewer 2 Report

Comments and Suggestions for Authors

Thanks for the revision.